# Dense Backpropagation Improves Routing for Sparsely-Gated Mixture-of-Experts

## Abstract

Mixture of Experts (MoE) pretraining is more scalable than dense Transformer pretraining, because MoEs learn to route inputs to a sparse set of their feedforward parameters. However, this means that MoEs only receive a sparse backward update, leading to problems such as router load imbalance where some experts receive more tokens than others. We present a lightweight approximation method that gives the MoE a dense gradient while only sparsely activating its parameters. A key insight into the design of our method is that at scale, many tokens not routed to a given expert may nonetheless lie in the span of tokens that were routed to that expert, allowing us to create an approximation for the expert output of that token from existing expert outputs. Our dense backpropagation outperforms standard TopK routing across multiple settings without significantly increasing runtime.

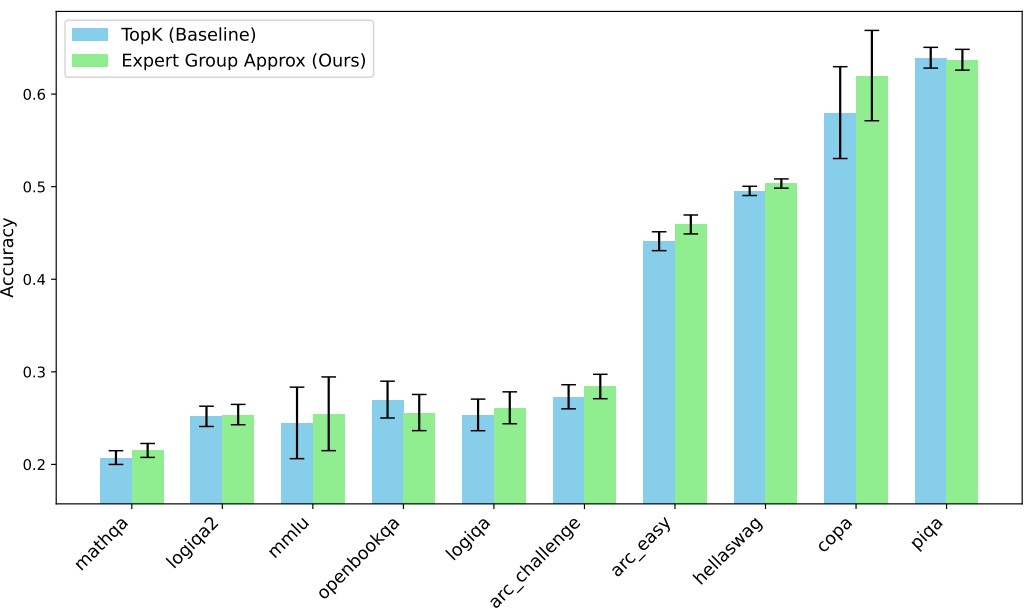

Figure 1: Our dense backpropagation method (green) improves over the baseline (blue) that uses Top-K routing across a range of standard benchmarks. See Section 4 for the experimental setup.

## 1 Introduction

Large-scale pretraining hinges on scaling up the number of parameters in a model, because models with more parameters are more sample-efficient and require less training to reach the same performance as smaller models (Kaplan et al., 2020; Hoffmann et al., 2022). The most common model architecture is a dense Transformer architecture (Vaswani et al., 2023) because its performance scales well with parameters and data. However, a sparsely activated Mixture-of-Experts (MoE) Transformer architecture (Shazeer et al., 2017) has been used by many industry deployments (Team et al.,

2024; xAI, 2024; Databricks, 2024; Jiang et al., 2024; Snowflake, 2024; DeepSeek-AI et al., 2024) because MoEs have been shown to scale even better than dense Transformers (Clark et al., 2022; Du et al., 2022; Lepikhin et al., 2020; Fedus et al., 2022). MoEs learn a *routing* function that selectively activates the TopK subset of their modules, or *experts*, most relevant to a given input. This conditionally sparse activation (Jacobs et al., 1991; Jordan & Jacobs, 1994) allows us to multiplicatively increase the model parameter count without significantly increasing the cost of training or inference.

The sparse router enables MoEs to scale, but it also presents a challenge, because the router does not receive a gradient update from experts that it does not activate, and may not learn to route a token to its appropriate expert. This may cause load imbalance— where a few experts are over-utilized — leading to inefficient training and resource usage (Zoph et al., 2022; Zhou et al., 2022).

**In this work** we propose a new router that can receive a dense gradient update from a sparse forward pass to address the instability issues arising from sparse routing. Our method adds minimal overhead, but improves on the common Top-K routing in both performance and load balance.

## 2 BACKGROUND & RELATED WORK

**MoEs.** The MoE layer replaces the feedforward networks (FFN) of transformers and consists of two components : **1)** $N$ FFNs (*experts*), $E_0(x), E_1(x), \ldots E_N(x)$ and **2)** a router that assigns tokens to experts. Each input to the MoE layer is processed by $K$ experts where $K < N$, and this is the source of sparsity in MoEs. The $K$ experts are chosen by the router, which is a learnable component that maps each token to a set of weights over the experts. The router performs a linear transformation $\mathbb{R}^{d_{\text{token}}} \to \mathbb{R}^N$ which produces logits; these are normalized using softmax, resulting in a probability distribution over the experts. With the router's linear transformation parameterized by a matrix $W$, we can represent the expert weights $\pi$ in the following way:

$$\pi \in \mathbb{R}^N = \text{Softmax}(Wx) \tag{1}$$

Once we have these expert weights, we apply a routing function to decide which of $K$ experts to route and process this token through. We consider TopK because it is the most popular.

**Top-K routing.** A standard method to select $K$ out of $N$ experts given the expert weights is to select the experts corresponding to the $K$ highest weights. Top-K routing (Fedus et al., 2022) passes the token to the $K$ selected experts and averages the expert outputs using these weights to produce the final output. Experts not selected by the Top-K routing function do not process the token, and this introduces sparsity in MoEs. By representing the $K$ chosen experts as the set $A$, we can express the output of the MoE layer as an average of expert outputs weighted by the router scores:

$$y = \Sigma_{i \in A} \pi_i E_i(x) \tag{2}$$

The expert weights serve two roles. They are used by the routing function to decide which of the $K$ experts to process a token through, and also provide the weights for combining the outputs of the expert. The Top-K routing scheme makes the MoE layer desirable for training large, compute-efficient neural networks. It allows models to be scaled up, by way of increasing the total number of experts, while keeping the compute per token constant (as it is a function of $K$ and not $N$).

**The Router Gradient.** Consider the gradient of the MoE layer's output $y$ with respect to the router parameters $W$. We express $y$ as a function of $W$ by combining Eq. (1) and Eq. (2). With the chain rule, we can backpropagate through this function by considering the gradient at each respective step:

$$\frac{\partial y}{\partial W} = \frac{\partial y}{\partial \pi} \frac{\partial \pi}{\partial W} \tag{3}$$

The first term in Eq. (3), $\frac{\partial y}{\partial \pi}$, is straightforward to compute because the steps in Eq. (1) are easily differentiable, as they consist of linear operations and activations. But Eq. (2) is not differentiable because Top-K expert selection transforms the continuous router weights $\pi \in \mathbb{R}^N$ into a discrete set of selected experts $A$ with $\binom{N}{K}$ possible values. One way to get around backpropagation of nondifferentiable operations is to use the straight-through estimator (Bengio et al., 2013), which treats the operator as the identity function. With straight-through we bypass the Top-K routing function and Eq. (2) becomes the dot product between $\pi$ and the vector of all $E_i(x)$ with the following gradient:

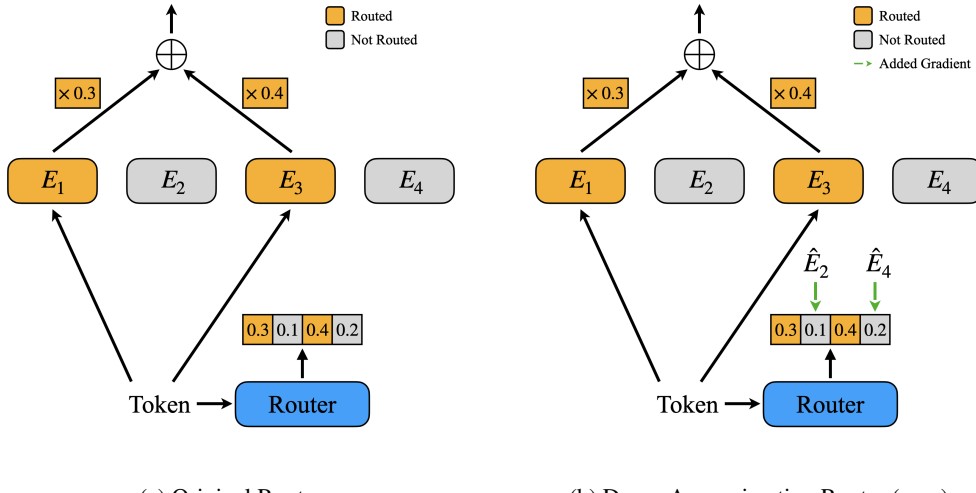

(a) Original Router         (b) Dense Approximation Router (ours)

Figure 2: **Overview of Routing with Dense Approximations**. The original mixture of experts router only receives gradients corresponding to experts the token is routed to, because there is no output from other experts. Our approach provides the router with a complete (dense) gradient, by approximating the activations of experts that a token is not routed to. As indicated by the dashed green arrows, the approximated gradients are not actually connected to the token in the computation graph; instead, they are artificially applied in the backward pass.

$$\frac{\partial y}{\partial \pi} = [E_1(x), \quad E_2(x) \quad \cdots \quad E_N(x)] \tag{4}$$

This gradient requires the output of *all* of the experts for that token. Passing a token through all the experts will destroy the sparsity of the MoE layer, ruining the scalability of this architecture. In this work, we develop methods for applying the straight-through estimator while maintaining the sparsity of the MoE layer by *approximating* the output of the experts not selected by Top-K routing.

**Related Works.** Previous work has tried to address the issue of routing in MoEs. Separate from Top-K is the Sinkhorn routing method (Clark et al., 2022). Fedus et al. (2022) propose an auxiliary loss that encourages load balancing. Dai et al. (2024) propose multiple additional auxiliary loss terms. Recently, Wang et al. (2024b) propose learning biases rather than an auxiliary load balancing loss. Even more recently, Phi-3.5-MoE (Abdin et al., 2024) uses SparseMixer (Liu et al., 2024; 2023), another estimator for $\partial y/\partial \pi$ not involving straight-through. We provide a comparison between our approach and SparseMixer later in Section 4.3. Our approach is to still use straight-through, but *approximate* these additional expert outputs. We now present our method.

## 3 DESIGNING DENSE BACKPROPAGATION

In this section we design a new backpropagation method that can send a dense gradient to the sparsely activated router and expert parameters without doing additional forward passes. In a standard MoE, both the embedding corresponding to expert $i$ in the routing layer (i.e. the $i$th row of the routing weight matrix) and the weights of expert $i$ receive no gradient update from a token $x$ if $x$ is not routed to expert $i$. This is because $E_i(x)$ is never computed, so it provides no upstream gradient. This corresponds to experts that are not in the top $K$ being omitted in Eq. (2). We apply an approximation $\hat{E}_i(x)$ as a substitute for the upstream gradient so that the router and expert weights can receive some non-zero signal corresponding to this expert. Then, the router can factor in outputs from all experts when learning to route each token, and the expert can be updated based on all tokens. Ultimately, instead of $K/N$ expert parameters being updated per token, every token will update more parameters in both the router and the experts. Updating the router should improve load balance and the routing distribution.

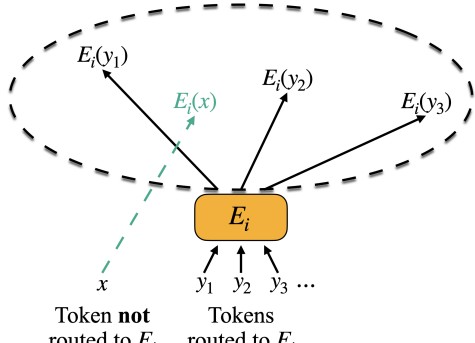

Figure 3: **Motivation for approximating expert outputs.** Consider an expert $E_i$. In every training batch, some fraction of tokens $Y$ will be routed to $E_i$. With a large training batch size, we expect more tokens $y_j$ to be routed to $E_i$. It is likely, then, that the outputs $E_i(y_j)$ will span the output space of this expert. Thus we expect that at the pretraining scale, a linear combination of expert outputs $\sum_j \alpha_j E_i(y_j)$ can approximate $E_i(x)$, for a token $x$ which was not routed to $E_i$.

### 3.1 Approximating Expert Activations With Dense Backpropagation

To approximate the dense gradient in Eq. (4), we must approximate $E_i(x)$ for every expert $i$ that a token $x$ was not passed to. Although we have no information about what the function $E_i$ looks like for $x$, when training with large token batch sizes it is very likely that we have outputs of $E_i$ for many other tokens. We hypothesize that although $E_i(x)$ is not computed for a given $x$, the output lies in the span of other tokens' activations for $E_i$. In other words, $\hat{E}_i(x) = \sum_j c_j E_i(x_j)$ for some other tokens $x_j$. Our approximation method relies on finding these relevant $x_j$ and the corresponding weights $c_j$ to develop an approximation for a token. We visualize this general approach in Fig. 3.

**Dense Backpropagation via Expert Group Approximation.** Our dense backpropagation method that we dub *expert group approximation* produces an estimator $\hat{E}_i(x)$ from the expert outputs of other relevant tokens. We apply a single approximation to a large group of tokens that were not routed to expert $i$. This is efficient, but it may not be a good approximation for any specific $x$. Instead, this is an estimator for the expert output across the entire batch - this is sufficient as we will only need an approximation for the batch gradient to update the router during training. In dense backprop, we select relevant $x_j$ and $c_j$ purely based on the router output.

This expert approximation is used solely to estimate the dense gradient in equation 4. We do not use this dense approximation in the forward pass, as that is inconsistent with the sparse forward pass we expect at inference. Additionally, applying this approximation at inference is infeasible because of the much smaller batch size. However, we do use this approximation to update the experts along with the router. As we will show, updating all the experts is a crucial component of our method, and something prior work has not done. In particular, Liu et al. (2023) focuses on the router gradient.

### 3.2 Notation on Expert Routing

Let $R(x)$ be the set of indices corresponding to the $K$ experts that a token $x$ is routed to. This can be thought of as the *routing decision* for $x$, based on the selected experts $A$ in Eq. (2). For example, in a top-k sparse mixture of experts layer with $N = 8$ experts and $K = 2$, $x$ routed to the first and last experts will have $R(x) = \{1, 8\}$. Note that $R(x)$ can have $\binom{N}{K}$ possible discrete outputs. With this notation, we can partition all tokens $X$ based on their routing decisions and denote $X_R$ as the subset of tokens routed to experts indexed by $R$. In the preceding example, the token routed to the first and last experts would belong to the set $X_{\{1,8\}}$. Some of our methods involve denoting whether a token was routed to a set of experts instead of its exact routing decision. We denote tokens routed to expert $i$ along with any other experts as $X_{\{i,\cdot\}}$. For example, $X_{\{1,8\}} = X_{\{1,\cdot\}} \bigcap X_{\{8,\cdot\}}$

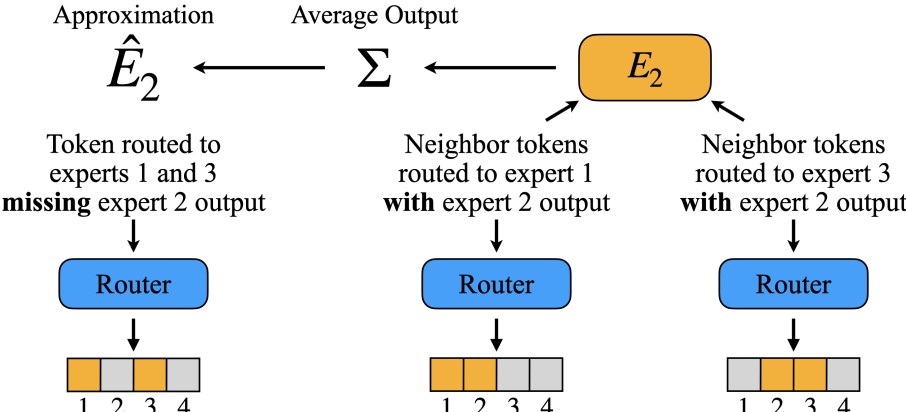

Figure 4: **Architecture of the Expert Group Approximation method.** In this example, we have 4 experts with $K = 2$. Consider all inputs routed to experts 1 and 3, characterized by the routing decision $R = \{1, 3\}$. As described in Figure 2b, we need to approximate these inputs' activations for all other experts. In approximating expert 2, for example, we collect all inputs $x'$ with a routing decision similar to $R$ specifically including expert 2: $R' = \{1, 2\}$ and $R' = \{2, 3\}$. In general there will be $K$ such adjacent groups. The aggregation of these inputs' activations for expert 2 is used to approximate expert 2 for all inputs routed to experts 1 and 3.

### 3.3 EXPERT GROUP APPROXIMATION

Our method is based on approximating the expert output $E_i(x)$ for a group of multiple tokens at once. For a token $x$, we want to approximate outputs of experts that $x$ was not routed to, i.e. $E_i(x)$ where $i \notin R(x)$. We hypothesize that tokens being routed to the same expert is a strong indicator of similarity between the tokens. This is because tokens routed to the same expert $E_i$ both yield high dot products with the embedding $W_i$ corresponding to that expert in the routing layer (see Eq. (1)). We develop an approximation for $E_i(x)$ by aggregating outputs of $E_i$ for tokens that were routed to both expert $i$, and other experts that $x$ was routed to. Formally, we consider an alternate routing decision $R' = \{i, j, \cdot\}, j \in R(x)$ that consists of one expert $j$ that $x$ is routed to, the expert $i$ we wish to approximate, and any other experts (if $K > 2$). Then, the adjacent token space $X_{R'}$ will consist of tokens that are very similar to $x$ by virtue of having similar routing decisions. Moreover, they will have an output for expert $i$, and we hypothesize that their outputs $\sum_{x' \in X_{R'}} E_i(x')$ will approximately represent $E_i(x)$. We can aggregate such outputs over all possible routing decisions:

$$\forall x \in X_R : \hat{E}_i(x) = \frac{1}{K} \sum_{j \in R} \frac{1}{|X_{\{i,j,\cdot\}}|} \sum_{x' \in X_{\{i,j,\cdot\}}} E_i(x') \tag{5}$$

We apply a single aggregate approximation for each routing decision to all tokens with that routing decision. In other words, each token belongs to $K$ "groups" corresponding to the $K$ experts it was routed to. Each expert has $N$ groups, and each group $j$ receives an approximation for expert $i$ based on tokens routed to experts $i, j$. This gives us $N^2$ total approximations. When we approximate expert $i$, a token receives one approximation from each group, scaled by $\frac{1}{K}$ to take the average. In Fig. 4 we visualize this method for $K = 2$. We can pick different constants for the number of activated experts and the number of groups, so our method is applicable to $K = 1$, but we keep these constants at $K = 2$ throughout the paper because it is a common choice across prior work.

As mentioned above, we use this approximation to update both the router and the experts. This is necessary to maintain consistency, as updating the router densely but the experts sparsely can cause them to diverge. We also omit the approximation from the forward pass to only apply it in the backward pass. Consider an expert approximation based on other token outputs $\hat{E}_i(x) = \sum_j c_j E_i(x_j)$ assuming $\sum_j c_j = 1$. We produce such approximations for all experts in $R(x)^C$ that

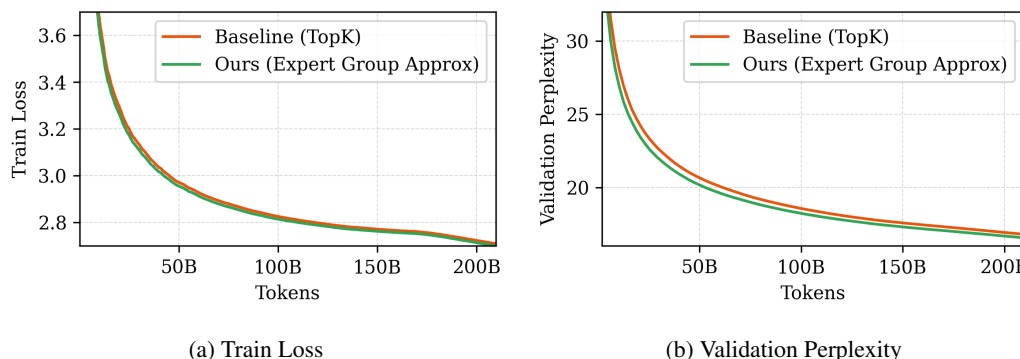

(a) Train Loss

(b) Validation Perplexity

Figure 5: **Training and Validation Results.** We train a $2B$ total parameter MoE with 32 fine-grained experts, using the standard Top-K=2 router ($470M$ active params) and our expert group approximation method, each on 200B tokens of the Fineweb dataset. Without incurring significant overhead, we improve over the baseline.

$x$ was not routed to: $y' = \sum_{i \in R(x)^C} \hat{E}_i(x)$. We update the original MoE output $y$ from Eq. (2):

$$y := y + y' - \mathrm{sg}(y') \tag{6}$$

where sg is the stop-gradient operator. As a result, our forward pass output is unchanged. During backpropagation, the expert approximations $\hat{E}_i(x)$ fill in the missing entries of the dense router gradient in equation 4. Moreover, the gradient to an expert $i$ from the token $x_j$ used to approximate $\hat{E}_i(x)$ is increased by the upstream gradient $\partial \hat{E}_i(x)$ times the weight $c_j$. In expectation, roughly $K/N$ tokens are routed to an expert $i$ so $(N - K)/N$ tokens will add gradients $\partial \hat{E}_i$ based on their approximations. Overall, this will lead to a dense gradient approximation for each expert since it will receive gradients as if it processed $K/N + (N - K)/N = N/N$ of all tokens instead of $K/N$. We all-reduce the approximation gradients across data parallel workers. This gives us a large sample size to estimate the dense gradient, regardless of the micro batch size on each worker.

## 4 EVALUATION

We train a fine-grained (DeepSeek-AI et al., 2024) MoE with 32 experts. The model has a hidden dim of 1024, with 2B total parameters. Each expert is a bottleneck MLP that has intermediate size 704. 470M parameters are activated when doing the standard $K = 2$ top-K routing (Zoph et al., 2022). We ablate the model architecture, hidden dim, number of total experts, and number of active experts through the course of the evaluation in Table 2 and Table 3. We train on 200B tokens from FineWeb (Penedo et al., 2024) with the Llama3 tokenizer (Dubey et al., 2024). We split it into train, validation, and test splits and report the validation perplexity. We defer other implementation details (learning rate schedule, initializations) to Appendix B.

### 4.1 MAIN RESULTS

**Main Result.** Our main result compares the Expert Group Approximation, which performs a dense update of the router weights by approximating the dense gradient, to baseline Top-K routing. In Fig. 5 we plot both training loss and validation set perplexity over the course of training. Our Expert Group Approximation method yields improvements over the baseline Top-K routing method throughout training, and this improvement is still present after training on 200B tokens. We further evaluate the performance of our Expert Group Approximation on standard benchmark tasks (that Penedo et al. (2024) notes are "high signal" for models trained on Fineweb) in Table 1 and find that our method provides a modest improvement over Top-K across multiple benchmarks.

Table 1: **Benchmarks.** Our method improves over the baseline across a set of benchmarks.

| | mathqa | logiqa2 | mmlu | openbookqa | logiqa | arc_challenge | arc_easy | hellaswag | copa | piqa | Average |
|---|---|---|---|---|---|---|---|---|---|---|---|
| Baseline | $20.7_{0.01}$ | $25.2_{0.01}$ | $24.5_{0.04}$ | $\mathbf{27.0}_{0.02}$ | $25.3_{0.02}$ | $27.3_{0.01}$ | $44.1_{0.01}$ | $49.5_{0.00}$ | $58.0_{0.05}$ | $\mathbf{63.9}_{0.01}$ | $36.6_{0.02}$ |
| Ours | $\mathbf{21.5}_{0.01}$ | $\mathbf{25.4}_{0.01}$ | $\mathbf{25.5}_{0.04}$ | $25.6_{0.02}$ | $\mathbf{26.1}_{0.02}$ | $\mathbf{28.4}_{0.01}$ | $\mathbf{45.9}_{0.01}$ | $\mathbf{50.3}_{0.00}$ | $\mathbf{62.0}_{0.05}$ | $63.7_{0.01}$ | $\mathbf{37.4}_{0.02}$ |
| Improvement | +0.8% | +0.2% | +1.0% | −1.4% | +0.8% | +1.1% | +1.8% | +0.8% | +4.0% | −0.2% | +0.9% |

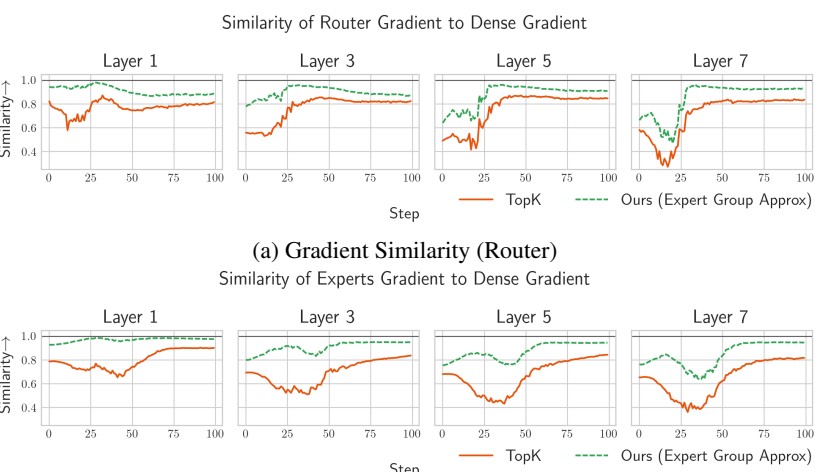

(a) Gradient Similarity (Router)

(b) Gradient Similarity (Experts)

Figure 6: **Approximation fidelity.** We train MoEs with 32 fine-grained experts, recording the similarity with the true dense gradient (for the router and the experts) for the standard Top-K=2 baseline and our method. The gradients returned from our method are significantly closer to the true dense gradient. While prior work such as Liu et al. (2023) only aimed to densely update the router, we also densely update the experts, and we see that we actually do a better job of updating the experts than the router. Because Liu et al. (2023) do not design their method to approximate the true dense gradient of the router, we do not report their gradient similarity here (although it is low, their method achieves routing imbalance and performance in an entirely different mechanism to ours). *Our method faithfully reconstructs the true dense gradient signals.*

## 4.2 EMPIRICAL ANALYSIS

**Gradient Approximation.** We verify that our method is indeed faithfully approximating the dense router gradient i.e. the gradient to the router if all experts were activated. We track the dense gradient by routing to all experts and backpropagating only on the MoE output (independent of the full forward pass). In Fig. 6a we compare this dense gradient to the actual router gradient for each of our approaches and the standard Top-K router, and similarly for the expert weights in Fig. 6b. We plot the cosine similarity between the dense gradient and our approximated gradient, where the relatively high similarities our methods achieve indicate we are more accurately approximating the dense gradient. We also observe a major difference between the gradient of the standard Top-K router and our approach in later layers. We record the gradient similarity from the start of training until the trends stabilize; if we were to extend these plots out, we would see the similarities stay relatively stable. This validates our motivation to try and approximate the true dense gradient.

**Load Balance.** One reason for our method's superior performance is in how it improves the routing distribution. Without a gradient signal for unactivated experts, top-K routing may not learn a balanced distribution across experts, and route more tokens to some experts than others. On the other hand, our method sends a signal to the router even for experts that a specific token was not routed to. The router now receives more information to make better decisions on how to route inputs. Load imbalancing leads to some experts processing a larger fraction of tokens than others, and thus having to learn a representation for more tokens. although all experts share the same number of parameters. Balancing expert load thus allows for more efficient utilization of MoE parameters.

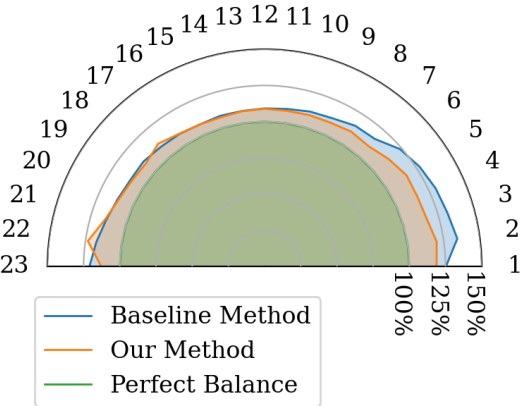

Figure 7: **Load Balancing using Expert Group Approximation**. We define maximum load imbalance at each layer as $\max_i(N \cdot f_i)$ where $f_i$ is the fraction of tokens routed to each of $N$ experts. The green ring indicates perfect balance, where each expert receives $1/N$ fraction of tokens; the outer ring indicates a maximum imbalance of $150\%$. We record maximum imbalance after training on 3 billion tokens. Our method improves load balance by sending a complete gradient to the router.

In Fig. 7 we validate that the baseline top-K ($K = 2$) routing has an "imbalanced load", as measured by the proportion of tokens being routed to different experts relative to the optimal balance (green line) of an even distribution of tokens across experts. Our method improves load balance, which may be one cause for improved performance and is of independent interest on its own because it may lead to more efficient inference (although we do not profile inference workloads in this work).

### 4.3 ABLATIONS

Table 2: **Method scaling.** Our method's training perplexity improvement after 10B tokens scales with the number of total experts, batch size, and number of active experts. The 8-expert MoE and 32-expert MoE both have $2B$ total parameters and use $K = 2$. However, in the 32-expert MoE, each MLP is a bottleneck layer. The 32-expert MoE therefore has just $470M$ active params, while the 8-expert MoE has $780M$ active params. We use the best number of experts (32) for the batch size scaling experiments, and we use the best batch size ($2^{21}$) for the expert scaling experiments. We also vary the number of active experts between $K = 2$ and $K = 4$ for the 32-expert MoE and find that our method provides consistent improvements across different values of $K$.

| | Num. Experts | | Batch Size | | | Active Experts | |
|---|---|---|---|---|---|---|---|
| | 8 | 32 | $2^{19}$ | $2^{20}$ | $2^{21}$ | 2 | 4 |
| Baseline | 22.03 | 22.23 | 27.54 | 24.51 | 22.23 | 22.23 | 21.54 |
| Ours | **21.80** | **21.91** | **27.51** | **24.37** | **21.91** | **21.91** | **21.22** |
| Improvement | 1.1% | **1.5%** | 0.1% | 0.6% | **1.5%** | **1.5%** | **1.5%** |

**Scaling.** In Table 2 we show that our method's improvement over the baseline increases as we scale up training. As we increase the number of experts from 8 in a standard MoE architecture, to 32 fine-grained experts, our improvement over the TopK baseline increases from $1.1\%$ to $1.5\%$. This is because the 32-expert MoE has *more sparsity*, and therefore the gap between the baseline and the true dense gradient (that we are trying to approximate with our method) is larger. Our method constructs an approximation for unactivated experts from the rest of the tokens in the batch, so as the total training batch size increases, the likelihood that a linear combination of other tokens can

reproduce these missing expert outputs increases, as shown in Fig. 3. As we increase the train batch size, our improvement over the baseline increases from $0.1\%$ at $2^{19}$ to $1.5\%$ at $2^{21}$. DeepSeek-AI et al. (2024) start with a batch size of $\approx 2^{23}$ and ramp up to a batch size of $\approx 2^{25}$, so the batch sizes we consider are still significantly smaller than those used to train the SOTA MoEs. *The improvement of our method over the baseline may be marginal at the current scale, but we believe that our method will continue to improve over the baseline as we scale up our models and batch sizes.*

Table 3: **Method variations.** We report perplexity after training an $8$-expert $2B$ total-parameter MoE for $10B$ tokens. The "Accurate" variation on our method weights tokens based on how likely they were to be routed to the expert we're constructing an approximation for. The "Viable" variation weights by the probability of the expert we're using the approximation for. The "Accurate" variation is the best version of our method, because it gives us more accurate approximations.

| Routing Method | Val. PPL ($K = 1$) | Improvement | Val. PPL ($K = 2$) | Improvement |
|:---:|:---:|:---:|:---:|:---:|
| Baseline | 23.91 | – | 23.16 | – |
| Expert Group Approx. | 23.62 | $0.29\%$ | 22.62 | $0.54\%$ |
| "Accurate" | **23.48** | **0.43%** | **22.50** | **0.66%** |
| "Viable" | 23.49 | $0.42\%$ | 22.63 | $0.53\%$ |

**Variations on the method.** Thus far we have presented results where the approximation we use for a group of tokens is just a simple average of the expert outputs for similar tokens. However, when we construct the approximation we can weight tokens based on their router scores. In Table 3 we compare two variations on our main method. As a reminder, in this method we construct a mask of shape $experts, experts$ for each token. The row is the expert that token was routed to, and the column is the expert we want an approximation for. When we take the product of this mask and the router scores, we can weight each row by the probability corresponding to the expert we want an approximation for, or weight each approximation by the probability for the expert we're using the approximation for. The former should give us more "accurate" approximations, because it will prioritize tokens that are more likely to be routed to the expert we want an approximation for. The latter should give us more "viable" approximations, because it weights by closeness to the space we're using the approximation for. We find that both variants improve over our original method, with the "accurate" variation proving to be the best across $K = 1$ and $K = 2$. *Stacking the "accurate" variant on top of our method would further improve our main results.*

**Scaling with Sparsity.** We present most results in this paper with $K = 2$ because Zoph et al. (2022) finds that it improves performance over $K = 1$ with minimal overhead, and thus multiple prior works use $K = 2$. However, we can see in Table 3 that our method is *also better* at $K = 1$. Virtually zero MoE papers train with $K = 1$; the closest is Sun et al. (2024), and they use a shared expert (DeepSeek-AI et al., 2024) alongside the $K = 1$ expert, so in fact 2 experts are active at each layer. We believe this can be a new capability that our method unlocks, which in turn will lead to even greater sparsity and therefore more efficient MoEs.

**Comparison to Sparsemixer.** Liu et al. (2023; 2024) use Sparsemixer, which estimates the true router gradient without straight-through. Our method is distinct in that we also provide gradient updates for the expert weights (as seen in Fig. 6b). Liu et al. (2024) note that Sparsemixer lags behind TopK (which our method always outperforms) for the first $0.5T$ tokens, likely due to the noise that Sparsemixer adds. In the $10B$ tokens that we were able to run Sparsemixer for, our method significantly outperforms it.

**Further Ablations.** We conduct further ablations on design choices in Appendix C.1.

## 4.4 Efficiency

Although our method outperforms the baseline in all settings we consider, to ensure a fair comparison it is crucial that our method does not significantly increase wall-clock time. In Table 4 we report the throughput in samples per second and TFLOPS of TopK, Sparsemixer, and our method while training a fine-grained MoE with 32 experts and $K = 2$ on a single GPU (for reproducibility). We find that the overhead of Sparsemixer and our method decreases as we increase the hidden size. The $2B$ model we train has $1024$ hidden size, 2048 hidden size is an $8B$ model, etc. As the hidden size increases, the proportion of time spent in the MLP matmuls increases, and the overhead of our model

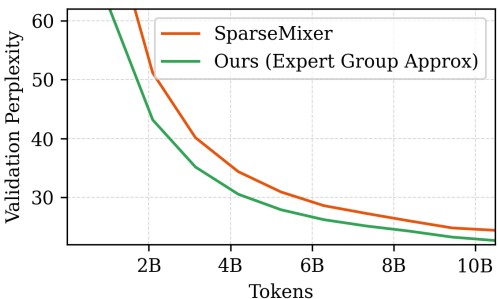
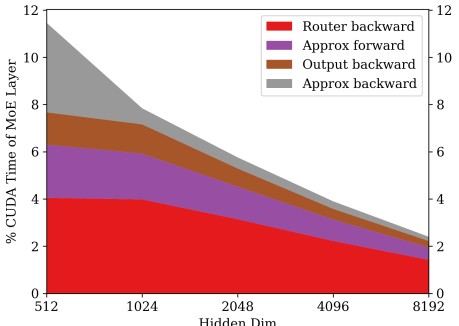

Figure 8: **Sparsemixer comparison.** We train a fine-grained MoE with 32 experts with our method and Sparsemixer (Liu et al., 2023). Our method, which outperforms the TopK baseline from the start of training, outperforms Sparsemixer. Liu et al. (2024) report that Sparsemixer requires many ($0.5T$) tokens to catch up to the TopK baseline.

Figure 9: **CUDA overhead scaling.** The overhead of our method, as measured in the percent of the CUDA time it occupies, decreases as we increase the hidden size of the model. Our main experiments are conducted with a hidden size of $1024$, resulting in $470M$ active parameters and $2B$ hidden parameters. A hidden size of $4096$ would be commonly trained model.

Table 4: **Throughput Comparison.** We compare the throughput between different routing methods across hidden dimensions. Throughput is measured in tokens per second (sequence length 2048) on a single GPU. Overhead is calculated relative to TopK routing.

| Hidden Dim | Tokens per second | | | Overhead vs TopK | | |
|---|---|---|---|---|---|---|
| | TopK | Sparsemixer | Ours | TopK | Sparsemixer | Ours |
| 1024 | 87,039 | 81,018 | 75,449 | - | -6.92% | -13.32% |
| 2048 | 12,288 | 11,469 | 11,790 | - | -6.67% | -4.05% |
| 4096 | 10,404 | 10,180 | 10,240 | - | -2.15% | -1.57% |

is no longer significant compared to the total MoE layer runtime. The same is true for Sparsemixer, although our method is slightly faster than Sparsemixer for hidden sizes 2048 and larger.

We further analyze this in Fig. 9, where we record the percent of CUDA time of the MoE layer that is spent in each part of our method. We develop Triton (Tillet et al., 2019) kernels to efficiently implement our method. Our code is currently open-source and will be linked here upon publication. For larger models, most of the overhead comes from the "Router backward" kernel where we compute the gradients to pass to update the router.

The overhead we report in Table 4 does not account for communication overhead. On our cluster, about half of the iteration time for the 1024 hidden size MoE is spent in NCCL operations when training on 8 nodes, and as we increase the number of nodes, the communication costs dominate following the trend of Wang et al. (2024a). Although performance varies dramatically across clusters, in our training setup our method adds near-zero overhead when training MoE with hidden size greater than 2048 (that is, more than 10B parameters).

## 5 DISCUSSION

We propose a training method for MoEs to improve load balancing and language modeling performance. By approximating the signal of a dense mixture-of-experts layer, the MoE router is able to learn a better distribution of routing inputs to different experts. This approximated dense signal unlocks the possibility for more sparse MoEs at training and inference time. Whereas typical Top-K routing would provide too sparse of a signal to learn a stable routing distribution, our method demonstrates significant improvements in load balancing in very sparse configurations. This unlocks better performance on standard language benchmarks, and has little overhead.

## 6 REPRODUCIBILITY STATEMENT

We provide detailed explanations of all our proposed methods in order to encourage reproducibility of our work: see Fig. 4 and Fig. 14. We use open-source LLM training libraries (Andonian et al., 2023; Gale et al., 2022) for implementing our experiments, and we train our models using a publicly available dataset (Penedo et al., 2024). Our methods are built on top of these open-source libraries, and we plan to release our code publicly in the future. We also provide the details of our experiments in Section 4, and further details on hyperparameters are included in Appendix B.

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

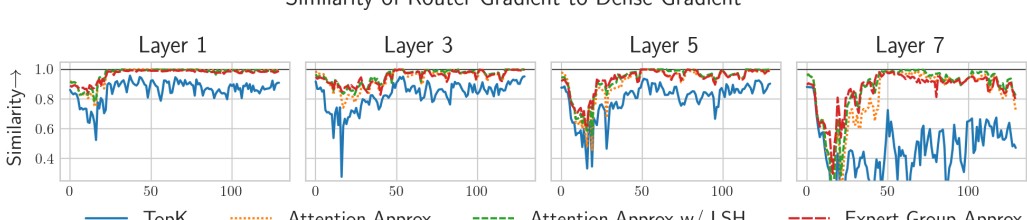

Figure 10: **Accuracy in approximating the dense router gradient for each approach**. We use a model with 8 experts and $K = 2$. We artificially compute the *dense gradient* of the output with respect to the router weights at each step by passing inputs through a dense mixture of experts layer, where all experts are selected. We do this independent of the actual forward pass computation, while using the same set of MoE parameters. The similarity between this dense gradient and the actual gradient propagated to the router indicates how well the router is learning from all experts. We plot this similarity with a standard Top-K router, and with each of our proposed router modifications. Our approaches are much more accurate and stable in approximating the dense router gradient.

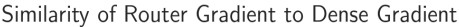

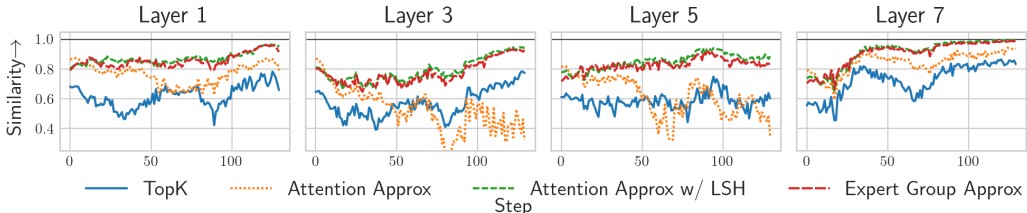

Figure 11: **Dense router gradient approximation accuracy with fine-grained experts**. We implement fine-grained experts as in DeepSeekMoE (DeepSeek-AI et al., 2024) to observe the behavior of our approximation methods across more experts while keeping parameter count fixed. In this example, the model now has 32 experts with $K = 2$. With more experts, it becomes increasingly hard to approximate the dense gradient, and the difference between our methods and the Top-K router is more apparent. Moreover, we can clearly compare the efficacy of each method and see that the attention approximation with LSH is the best. Note the average number of tokens per expert also decreases by a factor of $4$ as well, and we would expect even better performance in our approximations by scaling the local batch size per GPU, because we don't communicate tokens across ranks.

## A APPROXIMATION STATISTICS

### A.1 FURTHER GRADIENT ANALYSIS

The differences in our approaches become clear as we scale the model to become more sparse. We expand to $N = 32$ experts while maintaining $K = 2$ in Fig. 11 and find that it is more difficult to approximate the true dense router gradient. While all of our approaches sufficiently approximate the dense gradient with $N = 8$ experts, the performance gap between them is apparent with $N = 32$. The expert group approximation and LSH attention methods are significantly better than the direct attention method, and this is also consistent with our validation results in Table 6. This is likely due to the heuristics we apply to restrict our approximation to only the most relevant tokens: the expert group approximation requires tokens to have an expert in common, and LSH requires tokens to be similar. Moreover, the gap between our methods and Top-K is wider with 32 experts. We believe that in larger models with even more experts, our method will yield increasingly significant improvements over Top-K routing. We provide further analysis on our approximations in Appendix A. In Fig. 12 we provide an additional analysis of the gradient norm of our approximation compared to the dense gradient. This logging is also done with $N = 8$ and $K = 2$. The Top-K gradient has

Table 5: Our expert group approximation obtains the best validation perplexity after 20B tokens on a standard MoE architecture, achieving the same performance as $K = 3$ without activating an additional expert, meaning that we achieve the best performance-FLOPS tradeoff.

| Activated Experts | Routing Method | Validation Perplexity |
|---|---|---|
| $K = 1$ | Baseline | 19.61 |
| $K = 2$ | Baseline | 18.92 |
| $K = 3$ | Baseline | 18.56 |
| $K = 2$ | Expert Group Approx. (Ours) | **18.55** |

significantly lower norm than the dense gradient. Our methods closely approximate the dense gradient norm consistently; replicating both the direction and magnitude suggests that we are sufficiently approximating the dense gradient entirely.

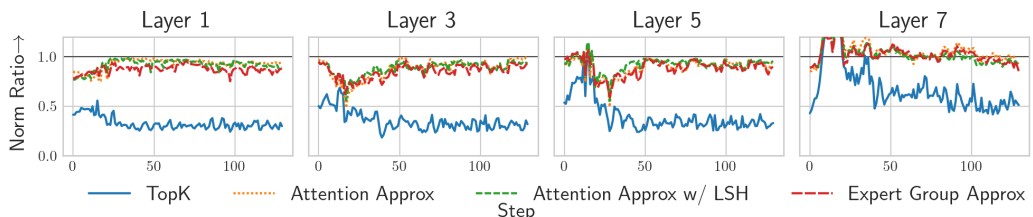

Figure 12: **Comparison of gradient norms relative to the dense gradient**. When computing the dense gradient, we also record its $L_2$ norm and log the ratio of this to the $L_2$ norms of the actual router gradients during training. Our methods produce router gradients with approximately the same magnitude. Along with the results showing strong cosine similarity, this suggests that we are almost perfectly approximating the dense gradient.

In Fig. 5 we presented results with a Deepseek-style finegrained MoE. In Table 5 we present results with a conventional architecture. This model has 2B hidden parameters and has 780M active parameters. Our method outperforms the baseline. As we will show, our method improves performance by improving load balance. Load balancing is especially important early on in training – this is where our method shows the largest improvement in Fig. 5. Improving the load balance of experts yields benefits akin to actually activating more experts. In Table 5 we find that our lightweight approximation method improves performance by a similar amount as activating an additional expert (that is, going from $K = 2$ to $K = 3$), without the additional computational overhead during training and inference of actually needing to use the parameters of a third expert. The choice of $K = 2$ in all experiments follows Zoph et al. (2022).

## B  HYPERPARAMETERS AND IMPLEMENTATION DETAILS

**Model Configuration.**  Both MoEs have 24 blocks and a hidden dimension of 1024. The Deepseek MoE has fine-grained experts. Each expert is a bottleneck MLP of shape 1024, 704). The conventional MoE has an expansion factor such that the intermediate size of the MLP is 2816. We use SwiGLU (Shazeer, 2020) MLPs following Llama (Touvron et al., 2023), 16 attention heads with dimension of 64, LayerNorm (Ba et al., 2016) and RoPE (Su et al., 2023). **Hyperparameters.**  We use the initialization from  Wang et al. (2022) for residual branch merge layers and the initialization from Nguyen & Salazar (2019) for all other layers. We use the AdamW optimizer (Loshchilov & Hutter, 2019). We use the modified cosine learning rate schedule from Ibrahim et al. (2024). We use a sequence length of 2048 and a global batch size of 1024, resulting in a global token batch size of $2^{21}$. We set the auxiliary loss (Fedus et al., 2022) to 0.01.

**Implementation.**  We train with the gpt-neox library (Andonian et al., 2023) integrated with Megablocks (Gale et al., 2022). The TFLOPS vary depending on the method and the number of

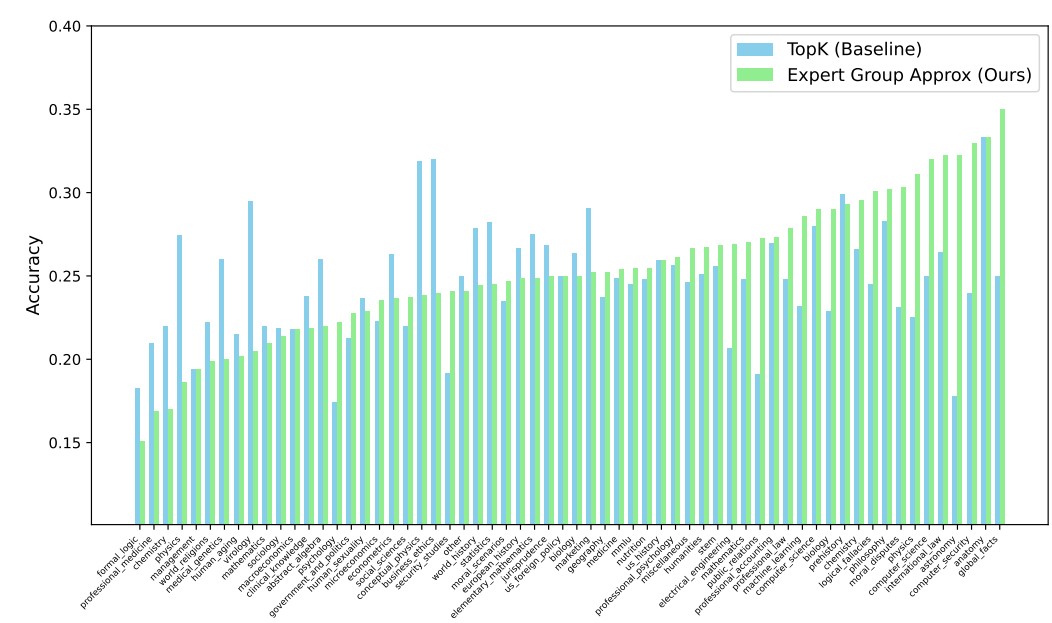

Figure 13: MMLU Scores of our method and the TopK Baseline.

experts chosen; for simplicity, we do not account for the router or the number of experts activated when reporting the TFLOPS, so that the number of flops we count in a forward and backward pass is the same as a dense model.

## C    TOKEN-SPECIFIC APPROXIMATION WITH ATTENTION

Our Expert Group Approximation computes an approximation, for each expert, for all tokens routed to it from each other expert, resulting in $N^2$ total approximations. However, we may want to actually compute an approximation for specific tokens. Consider tokens belonging to the set $x \in X^C_{\{i,\cdot\}}$, i.e. tokens *not* routed to expert $i$. We want to approximate $E_i(x)$ for such $x$. At a high level, we want to search for similar tokens to $x$, select their expert outputs $E_i(x_j)$, and aggregate these outputs as a weighted linear combination. This problem can be solved using attention. We want to query using all tokens *not* routed to expert $i$, i.e. $X^C_{\{i,\cdot\}}$. The keys will correspond to tokens that *were* routed to expert $i$, i.e. $X_{\{i,\cdot\}}$. And the values will be the expert outputs of these relevant tokens. Fig. 14a (left) outlines how we compute an approximation using multi-head attention, where each head corresponds to approximating for a single expert.

**Sparse Attention with LSH.** Computing attention across all tokens on a GPU is computationally expensive, and we do not need attention scores for *all* the tokens to compute the approximation, just for the most similar tokens to $x$. With a block-sparse attention mask, we can greatly reduce the attention computation, especially when most of the computed scores would be redundant. In Fig. 14b we outline our attention approximation that uses locality-sensitive hashing (LSH) to group tokens into buckets, with a high probability that the nearest neighbors to a token will lie in the same bucket. The attention mask now has an additional condition: the query index $q$ and key index $k$ must correspond to tokens in the same bucket. We sort the QKV into groups based on their assigned buckets to encourage a block-diagonal attention mask, and verify that this sparsity reduces the runtime of our attention approximation. Note that it is possible that some tokens receive no approximation because there are no keys to query in the bucket. In this case, we set the approximation to $0$.

Despite the reduction in computation from using a sparse attention mask, implementing this method is still computationally infeasible. We cannot use kernel based implementations like FlashAttention because our attention inputs are the high-dimensional model embeddings instead of the low-dimensional embeddings that FlashAttention expects. We believe future works expanding efficient

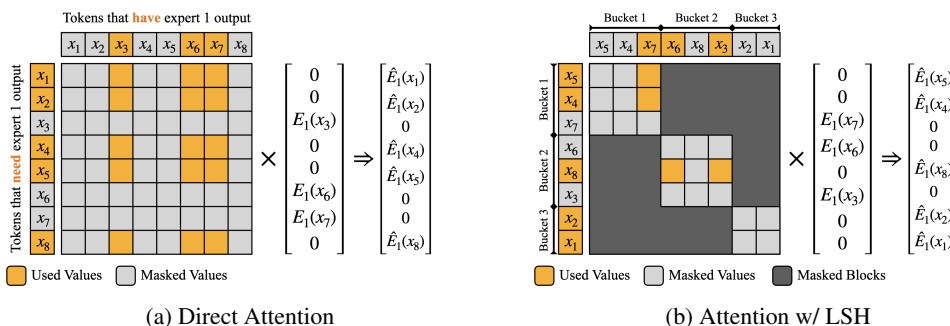

(a) Direct Attention          (b) Attention w/ LSH

Figure 14: **Attention scores of direct and LSH attention methods**. For each expert, we define an attention head that uses queries corresponding to inputs not routed to the expert, and keys corresponding to inputs routed to the expert. Grey entries denote queries and keys that do not meet this criteria, and whose attention scores are masked out. We multiply the attention scores of each head with the values, which are expert outputs of tokens routed to that attention head's expert. This implementation is common to both the direct and LSH attention method. In the latter, we further optimize the attention calculation by sorting inputs into buckets based on cosine similarity. This creates a block-sparse attention map, allowing kernels to skip most of the attention computation.

attention implementations to higher dimensions could make this token-wise approximation method more feasible.

### C.1 ABLATING DESIGN CHOICES

We ablate our main design choices and show that our main method does not compromise throughput.

**Comparing Different Approximation Methods.** We use the Expert Group Approximation method for our main results because it is lightweight, easy to implement, and provides good performance. However, the other two methods we consider also outperform the top-K ($K = 2$) baseline. Indeed, as we showed in Fig. 11, the Attention+LSH method seems to obtain a better approximation of the true dense gradient. The primary reason why we report our main results with Expert Grouping is because the Expert Group Approximation method requires no additional memory overhead. This allows us to use larger microbatches, and therefore there are more tokens on each GPU that we can use for the approximation. In Table 6 we find that even with a microbatchsize $4\times$ smaller than that of the Expert Group method, the Attention+LSH method is competitive.

| Routing Method | Microbatchsize | Validation Perplexity |
|:---:|:---:|:---:|
| Attention | 4 | 18.72 |
| Attention+LSH | 4 | 18.64 |
| Expert Group | 16 | 18.55 |
| Baseline | 16 | 18.92 |

Table 6: Comparison of routing methods after training on 20B tokens.

