# OpenReview forum: "Dense Backpropagation Improves Routing for Sparsely-Gated Mixture-of-Experts"
_ICLR.cc/2025/Conference — Submitted to ICLR 2025_

### Official Review · Reviewer_XWKc · 2024-10-20

**Soundness:** 3
**Presentation:** 2
**Contribution:** 3
**Rating:** 6
**Confidence:** 2

**Summary:**

The paper proposes a novel method to improve routing in Mixture of Experts (MoE) by approximating the oracle dense gradient to update the entire router for each token. The authors introduce a lightweight group averaging method, allowing MoEs to receive dense gradient updates during backpropagation without activating more experts in the forward pass. The proposed method outperforms standard Top-K routing across multiple benchmarks, reduces load imbalance, and maintains runtime efficiency. Alternative approximation methods like attention and LSH attention are also explored.

**Strengths:**

1. *Improved Performance*: The proposed Expert Group Approximation method achieves better results than standard Top-K routing in terms of training loss, validation perplexity, and downstream accuracy.
2. *Lightweight*: The method is computationally efficient, introducing minimal overhead while maintaining strong performance.
3. *Enhanced Load Balancing*: The method approximates the dense gradient well, leading to better load balancing than standard Top-K routing.
4. The method is verified on a relatively large model of 2B total parameters (470M active), and a relatively large training tokens of 200B. Ablations on various approximation heuristics are also presented, providing insight into the design choices.

**Weaknesses:**

1. *Motivation Clarification*: The authors propose using Expert Group Approximation to estimate the straight-through estimator (STE), suggesting that the dense gradient is superior. However, it is not clearly established that the sparsely activated MoEs trained with this STE-based dense gradient will perform better than those trained with sparse gradients. A direct experiment validating this assumption is needed to strengthen the motivation.
2. *Unclear Methodology Presentation*: The method is initially presented as focusing on approximating the router gradient during backpropagation, but in Section 4.2, it is revealed that applying the method only in the backward pass results in poor performance. Both the forward and backward passes need modification for the method to work effectively, which is fundamentally different. Introducing this crucial detail earlier would prevent confusion and improve clarity.
3. *Modest improvement*: The improvement in validation perplexity (from 18.92 to 18.55, approximately 2%) is relatively small. Considering the slight drop in efficiency (97.7% speed) and the limited experimentation on only two settings, the practical significance of this improvement could be questioned.
4. *Dependence on large micro batch size*: The methods need a large micro batch size to achieve good approximations, which is a limitation when training large-scale models with limited VRAM.

**Questions:**

1. The proposed Expert Group Approximation seems to be carefully designed for $K=2$. How does it perform with $K=1$ or $K>2$?
2. Does the method scale effectively with an increasing number of experts? Would it perform better in more sparse configurations with a higher number of experts?
3. The approximation of each token seems to involve activations or activation gradients of future tokens. Does this preserve causality in autoregressive models?

---

> ### Author Response · Authors · 2024-11-26
> **Author Response**
>
> Dear Reviewer XWKc,
>
> Thank you for the constructive comments, and for appreciating the strengths of our evaluation. In this response, we want to outline how we have addressed your concerns in the revised paper.
>
> > W1: Motivation Clarification
>
> Our method is motivated by previous works like SparseMixer that show approximating the dense gradient leads to improvements in training performance and stability. We develop a new estimator using the straight-through, and in Figure 1, Table 1 we demonstrate our improvements upon the baseline method as reported on a set of benchmarks. We include our improvement over the baseline through Section 4 (Evaluation).
>
> > W2: Unclear Methodology
>
> We recognize that our methodology was not clearly explained with regard to using the approximation in the forward pass. To clarify, our method does not use the approximation in the forward pass. We experimented with this and found the performance was better when only using the approximation in the backward pass. To answer your third question, this does not affect the causality of the language model, as the approximation information is not passed between tokens in the forward pass.
>
> > W3: Modest Improvement
>
> Thank you for your comments. You pointed out that while our lightweight method improves performance and load balancing, one weakness is the improvement being modest compared to the overhead increase. We have discussed how the improvement of our method over the baseline increases as we scale up training in Table 2. We also did a deep dive into the efficiency of our method in Section 4.4 “Efficiency”, with Figure 9 and Table 4. We believe that at scale, the improvement of our method is greater than the additional overhead, and also that the overhead can be further reduced by kernel specialists (which we are not).
>
> > W4: Dependence on large micro batch size
>
> We only use this approximation to update gradients in the backward pass; moreover, we all-reduce the approximation gradients across data parallel workers. This gives us a large sample size of token gradients to estimate the dense gradient, regardless of the micro batch size on each worker. As a result, our method does not depend on large micro batches to produce a good approximation. We will add an ablation over the micro batch size to validate this by the end of the author response period.
>
> > Q1: How does Expert Group Approximation perform for K=1 or K>2?
>
> To answer your question on values of K, we experimented with multiple of K and consistently observed an improvement in training. We report K=4 in Table 2 and K=1 in Table 3, and find that we still outperform the baseline with these other values of K.

---

> > ### Comment · Reviewer_XWKc · 2024-11-28
> >
> > Thank you for your reply. The revised manuscript and the clarifications have addressed most of my concerns. However, my primary concern regarding the motivation remains unresolved. I strongly believe that a STE-based dense-trained baseline is necessary for a comprehensive evaluation.
> >
> > - The authors state that the motivation is inspired by SparseMixer. However, SparseMixer's theoretical foundation is built on Gumbel-Softmax sampling, whereas this work employs deterministic TopK selection.
> > - SparseMixer formulates STE as a first-order approximation, while SparseMixer (or its v2) extends to second- or third-order approximations.
> > - Most results in the manuscript focus on perplexity and accuracy, which are valuable but insufficient to explain _why_ this method works. The similarity results with the STE-based dense gradient offer the only empirical insight into this question, but they are limited: the performance of using this STE-based dense gradient directly has not been demonstrated.
> > - A STE-based dense-trained baseline could serve as an upper bound for this method and provide a clearer perspective on its effectiveness.
> > - Additionally, I encourage the authors to provide deeper theoretical or empirical evidence to clarify _why_ this method works.
> >
> > I will consider raising my score if this issue is sufficiently addressed.

---

> > > ### Author Response · Authors · 2024-11-28
> > >
> > > Thank you for the detailed response! We are currently running the STE-based dense-trained baseline that you have suggested to provide an upper bound for the method. We plan to frame this result as the following:
> > >
> > > > The STE-based dense-trained baseline activates all the experts. Activating all the experts increases the computation by {this many times} and improves performance by {this much}, whereas our method does not significantly increase runtime and improves performance by {that much}. Therefore, our method can recoup a sizeable portion of the gains of the STE-based dense-trained baseline without significantly increasing overhead.
> > >
> > > If this result sounds good to you, we should have the numbers for 10B tokens by the end of the author response period. As with the rest of our ablations on 10B tokens, we will also queue a job for 200B tokens that should be done in time for the camera ready. We believe this will be the main result that can encourage you to revise your recommendation of our paper, but also provide responses to your other points below. We will incorporate this discussion into the camera ready version of the paper (since we can no longer edit the OpenReview PDF).
> > >
> > > > SparseMixer
> > >
> > > We agree with everything you have said. Instead of saying that our motivation is inspired by SparseMixer, we will instead say that our objective is inspired by SparseMixer, since the theoretical foundation of SparseMixer is not used in our work. Our work is trying to approximate the STE-based dense gradient for the router weights and expert weights, whereas SparseMixer is trying to approximate what is usually referred to as the policy gradient in RL to update the router weights. In principle, our method can actually be *combined* with SparseMixer by using SparseMixer for the router and only using our method to update the expert weights. We are currently running this ablation and it should be done by the end of the author response period.
> > >
> > > > The only empirical insight into why this method works is the similarity results with the STE-based dense gradient
> > >
> > > We also report the load balance results for the baseline and our method and show that our method achieves better load balance, which we believe is one element of why the method works.
> > >
> > > > Deeper theoretical or empirical evidence to clarify why the method works
> > >
> > > Thank you; we agree that we need to provide more evidence for why the method works, because the goal of a pretraining methods paper like ours is to derisk the prospect of using the method for researchers. We are working on additional empirical analysis in the flavor of the load balancing results, such as computing vocabulary specialization matrices.
> > >
> > > One intuition for why the method works is simply based on scaling laws. Dense training updates every parameter, while Sparse MoE training only updates K/N parameters. If we think about Chinchilla-optimal training, a hidden factor in how many tokens we should train on for each parameter in our model is how many parameters are actually updated. This argument is made by [1]:
> > >
> > > > MoE models need longer training to perform competitively but scale better after reaching that point. This should not be surprising since dense models use all parameters on each token contrary to MoE, which gains a computational advantage by activating only a subset of them.
> > >
> > > Our method can be seen as an approximation of STE-based MoE training where we activate all the experts, therefore approximating the computational advantage of using all parameters on each token, without actually incurring the corresponding compute overhead.
> > >
> > > [1] Krajewski, Jakub, et al. "Scaling laws for fine-grained mixture of experts." arXiv preprint arXiv:2402.07871 (2024).

---

> ### Author Response · Authors · 2024-12-02
>
> Dear Reviewer XWKc,
>
> We are following up with some initial results of the comparison between the STE-based dense-trained baseline and our method. We train an 8-expert MoE with 2B hidden parameters.
>
> After 9B tokens, we have:
>
> | Activated Experts | Method | PPL  |
> |-------------------|--------|------|
> | 2                 | TopK   | 22.06|
> | 3                 | TopK   | 21.38|
> | 8                 | TopK   | 21.08|
> | 2                 | **Ours**   | 21.38|
>
> These results corroborate our existing results in Table 5, where we find that using our method yields similar improvements to activating another expert, without the added compute cost of actually activating another expert. The experiments to run these for 20B tokens are currently running and we expect we may be able to update the results tomorrow.
>
> We will put these fleshed-out results into the camera ready. We believe that this STE-trained dense baseline serves as an upper bound for the performance of our method. This makes the intuition for our method clearer: we want to approximate what happens if we activate all the experts, because we know this will increase performance, and our goal is computational efficiency.
>
> We also have initial results for an ablation of our method where we use Sparsemixer for the router, and only use our method to provide a dense gradient for the experts. This seems to improve over just using Sparsemixer, and while it lags behind using our method without Sparsemixer, this may be expected because Sparsemixer adds noise and therefore will not be better until about 0.5T tokens [1]. We train an MoE with 8 experts and 2B hidden parameters, activating K=2 experts, for 10B tokens.
>
> | Method | PPL  |
> |-------------------|--------|
> | Sparsemixer (Router)   | 24.23 |
> | Sparsemixer (Router) + Ours (Experts) | 23.85 |
> | Ours (Experts) | 22.76 |
> | Ours (Router + Experts) | **22.41** |
>
> We believe this shows that our method is actually compatible with Sparsemixer, further bolstering the case that our method can be a valuable addition to MoE pretraining.
>
> [1] GRIN: Gradient Informed MoE. Liu et al. https://arxiv.org/abs/2409.12136

---

> > ### Comment · Reviewer_XWKc · 2024-12-03
> >
> > Thank you for the additional experiments. However, there seems to be some misunderstanding regarding the "STE-based dense-trained baseline". This does not simply involve activating all experts. Instead, it should sparsely activate 2 out of 8 experts during the forward pass and employ the STE-estimated dense gradient—precisely what your method further approximates—in the backward pass.
> >
> > Overall, the paper offers a reasonable approach to approximating STE-based dense gradients with minimal overhead. However, I remain skeptical about whether and to what extent the STE-estimated dense gradients can effectively guide the sparse forward results. Does the method perform well because it closely approximates STE, or is its success primarily due to increased gradient norms, which might behave similarly to using a higher learning rate?
> >
> > Since the performance and scaling behavior look promising, I’ve decided to raise my rating but lower my confidence.

---

> > > ### Author Response · Authors · 2024-12-03
> > >
> > > Ah, we apologize for the misunderstanding. We will run this baseline, where we activate the same number of experts and only change the backward pass, and provide the results in a response shortly. Thank you for engaging in the discussion period.

---

> > > > ### Author Response · Authors · 2024-12-03
> > > >
> > > > > Does the method perform well ... primarily due to increased gradient norms, which might behave similarly to using a higher learning rate?
> > > >
> > > > Thank you for this insightful question. We are aware that many methods will have a similar effect to increasing the learning rate, and to make sure this isn't the reason behind our method's success, we made sure to tune the baseline. We have tested learning rates between 1e-4 and 1e-3 for the baseline and used the largest stable rate, so a higher learning rate would not improve baseline performance. While the dense gradient (whether from activating all experts or approximating with our method) does have a larger norm, we use gradient clipping in both cases, preventing the larger gradient from effectively increasing the learning rate. We will include a table with the results of our hyperparameter sweep in the camera ready.

---

### Official Review · Reviewer_GiJM · 2024-11-02

**Soundness:** 2
**Presentation:** 1
**Contribution:** 1
**Rating:** 3
**Confidence:** 4

**Summary:**

This work proposes a dense approximation of the top K's gradient in SMoE. The method centralizes around the assumption that the output of an inactivated expert can be approximated by the outputs of other tokens routed to that expert. From there, the authors propose an expert group approximation to identify the relevant tokens given the current token and the inactivated experts. The experiments on a DeepSeek MoE and a conventional MoE show that the proposed method can perform better than the standard top K routing.

**Strengths:**

- Find a dense approximation of the top K routing is an important research topic.
- The proposed method show encouraging performances on some benchmarks.

**Weaknesses:**

- **The motivation of this work is not empirically validated** - It is unclear if the key motivation that an inactivated expert output can be reliably approximated by the activations of some other tokens. I think it is crucial to validate it on the models trained.
- **Poor presentation of the experimental results** - The authors implemented two SMoE models: a DeepSeek MoE and a conventional MoE. However, in many of the results, it is unclear which model is being reported, e.g. Figure 1, 7, 11, and Table 3,4. Overall, Section 4 is very confusing because of the inconsistent presentation of the results of the two models considered (e.g. validation perplexity for DeepSeek MoE as a plot in Figure 6 while it is a table for the conventional MoE in Table 4), some figures (7, 8) and tables (2, 3, 4) do not mention which model is being reported, and are not placed immediately where they are first mentioned.
- **Lack of baselines** - Dense approximations for the Top K operator is extensively studied in the literature, even the authors mentioned several relevant strategies (L140-146). Yet, the proposed method is only compared with the standard Top K baseline, ignoring all other baselines. It is important to show that the proposed method can show some advantages over state-of-the-art strategies such as SparseMixer.
- **Significance of the results** - Nowadays large scale LLMs, especially MoEs, are interested in the zero-shot evaluation settings. Yet these results are only briefly showed in Figure 1 and 11. What are the numbers on these benchmarks and are the improvements significant over strong baselines.
-  **Some claims are incorrect** - In L20, the authors claimed that the propose method does not "increase runtime", which is not entirely correct given that in L66-67 the authors stated that the method incurs "minimal overhead", and the throughput reported in Table 4. Moreover, the attention approximation clearly introduces more overhead.
- **Expert group approximation** - The experiment present in L479-496 may contradict the proposed method. The proposed "more accurate" approximation performs worse than the original method, suggesting that either the original idea may not be rigorous, or the proposed approximation is not more accurate. The authors didn't fully investigate this result, making this work seems incomplete and lack rigorousness.

Overall, despite some encouraging results, this work is still in-progress, both the motivations and experiments are fully investigated. I think the current manuscript is not ready for publication in its current stage.

**Questions:**

- Does the main motivation of the proposed method actually hold in practice?
- Can you present a more rigorous investigation of the results in L479-496?
- Can the propose method outperform some state-of-the-art strategies on zero-shot benchmarks?
- Please carefully revise Section 4 and make the claims regarding the complexity more accurate.

---

> ### Author Response · Authors · 2024-11-26
> **Author Response**
>
> Dear Reviewer GiJM,
>
> Thank you for your very useful comments, and for appreciating the importance of the problem we’ve chosen to study. In this response, we want to outline how we have addressed your concerns in the revised paper.
>
> > W1: Unclear how good our approximation is
>
> Thank you for your valuable feedback. We want to clarify that our goal is not to approximate the individual expert activation, but the dense gradient during the backward pass. We mentioned using the approximation in the forward pass in the paper, but after later experiments found that this was not optimal. We have edited the paper to reflect this in lines 198-201. In Figure 6 we demonstrate that we are very accurately approximating this dense gradient using the gradients of other tokens. Thus, the motivation of our paper does hold in practice when we train an MoE with the dense gradient approximation.
>
> > W2: Poor presentation of experimental results
>
> Thank you, we have rewritten the evaluation section. We now clearly state which method is being reported in each of our figures.
>
> > W3: No comparison to SparseMixer
>
> Thank you for the references. We have added a comparison to SparseMixer in Figure 8. We reproduce the text here for convenience:
>
> > \citet{liu2023sparsebackpropagationmoetraining, liu2024gringradientinformedmoe} use Sparsemixer, which estimates the true router gradient without straight-through. Our method is distinct in that we also provide gradient updates for the expert weights(as seen in Figure 8). \citet{liu2024gringradientinformedmoe} note that Sparsemixer lags behind TopK~(which our method always outperforms) for the first 0.5T tokens, likely due to the noise that Sparsemixer adds.
> Unfortunately, we have not been able to invest 0.5T tokens into this baseline comparison, but we are currently running this and expect to have it done for the camera ready.
> In the 10B tokens that we were able to run Sparsemixer for, our method significantly outperforms it.
>
> > W4: Significance of the results
>
> We have added Table 1 with precise numbers comparing the benchmark performance of our method and the baseline. Our baseline itself is a DeepSeek MoE with tuned coefficients for auxiliary loss and tuned LR, so we feel that we are beating a strong baseline. For the camera ready, we will add a row with the Sparsemixer results on these benchmarks as well.
>
> > W5: Runtime overhead
>
> Thank you, we agree that we were not clear about the runtime overhead of our method. We have added Section 4.4 “Efficiency” doing a deep dive into this, with Figure 9 and Table 4. We have updated “does not increase runtime” to “does not significantly increase runtime”.
> Our attention method was proposed as another alternative for approximating the dense gradient, but after further experiments we found this to be unreasonably efficient. This section is now entirely removed from the main text and moved to the appendix.
>
> > W6: Expert group approximation
>
> Since submitting the paper, we identified and fixed an error in our “accurate” and “viable” approximations. We have added new results for this in Table 3, and discuss it in Lines 450-462.
>
> > Q1: Does the main motivation hold in practice?
>
> We believe our gradient approximation similarity figure (Figure 6) confirms that our motivation does hold in practice, because our approximated gradient is more similar to the true dense gradient than the baseline TopK gradient is to the true dense gradient.
>
> > Q2: More rigorous investigation of "accurate" and "viable" approximations?
>
> We have now presented a more rigorous investigation in Table 3, and Lines 450-462.
>
> > Q3: Do we beat the SOTA on zero-shot benchmarks?
>
> We outperform a fine-grained TopK MoE trained with auxiliary loss, which is the setup used by SOTA MoEs including DeepSeek, on a range of benchmarks in Table 1.
>
> > Q4: Please make the claims regarding the complexity more accurate.
>
> We have carefully revised Section 4 per your guidance, and spun off the efficiency discussion into Section 4.4, backed by Figure 9 and Table 4.
>
> We hope that we have addressed your concerns and that the revisions made to the paper answer your questions.

---

> > ### Author Response · Authors · 2024-12-02
> >
> > Dear Reviewer GiJM,
> >
> > As today is the final day for reviewers to ask questions to the authors, we are following up to ask whether you have any remaining questions that we can answer about the range of new experiments we provided during the rebuttal period, or about the changes we made to the paper to address your comments. We would also like to draw your attention to the response to Reviewer XWKc, where we provided more results showing that our dense approximation can provide similar benefits to activating another expert without increasing compute cost significantly. We also provided results showing that our method is compatible with Sparsemixer and actually improves it. This is because Sparsemixer focuses on the router update, whereas our method can update both the router and the experts with a dense gradient, or some combination thereof.

---

### Official Review · Reviewer_3TBL · 2024-11-02

**Soundness:** 2
**Presentation:** 3
**Contribution:** 2
**Rating:** 5
**Confidence:** 4

**Summary:**

This paper introduces a novel method to enhance the performance of MoE models, which are crucial for large-scale language model pretraining due to their scalability advantages over dense Transformers. MoEs route input tokens to a sparse set of experts, leading to sparse backward updates and potential load imbalance issues. The proposed method addresses this by providing a dense gradient to the MoE router while maintaining sparse activation.

The key idea is that at scale, many tokens not routed to a specific expert might still be similar to those routed to that expert. This allows for approximating the expert output for unrouted tokens based on existing expert outputs. The authors propose three approximation methods: Expert Group Approximation, Attention Approximation, and Attention Approximation with LSH. Their experimental results demonstrate that Dense Backpropagation, particularly the Expert Group Approximation method, outperforms standard Top-K routing in both performance and load balance across various MoE configurations without increasing runtime.

**Strengths:**

The paper presents a novel approach to address the limitations of sparse backpropagation in MoEs. While previous work has focused on auxiliary loss functions or alternative routing methods, this work introduces the concept of approximating expert activations to provide a dense gradient signal. This is a creative solution that hasn't been explored in prior work.

The paper is well-written and easy to follow. The authors provide a clear explanation of their method, including visualizations and detailed descriptions. The notation is consistent, and the overall structure of the paper is logical and coherent.

By improving the load balancing and performance of MoEs, Dense Backpropagation can lead to more efficient training and resource utilization. This is particularly important as language models continue to grow in size and complexity.

**Weaknesses:**

The evaluation has been primarily focusing on perplexity evaluation. The review feel a more thorough evaluation on scoring and sampling tasks could significantly improve paper's quality. The authors acknowledge that their evaluation is limited in terms of the number of benchmark results, training tokens, and the size of the MoEs. Expanding the evaluation to include more diverse and challenging tasks, as well as larger models and datasets, would further strengthen the paper's conclusions.

While the paper focuses on the training benefits of Dense Backpropagation, it doesn't analyze its impact on inference workloads. Evaluating the method's efficiency and performance during inference would provide a more complete picture of its practical implications.

The paper primarily focuses on the Expert Group Approximation method due to its efficiency and ease of implementation. A more in-depth exploration of the other two approximation methods, Attention Approximation and Attention Approximation with LSH, could reveal further potential for improvement. If the other two methods are not useful at all. It is better to not mention the other two methods.

The paper lacks a comparison with more recent work in MoE routing. A few related work that can be compared:
- "Mixture-of-Experts with Expert Choice Routing" by Zhou et al. (2022)
- "DeepSeek-MoE: Towards Ultimate Expert Specialization in Mixture-of-Experts Language Models" by Dai et al. (2024)
- Gradient-Informed MoE (GRIN)
- SparseMixer

**Questions:**

1. The paper consistently uses K=2 for Top-K routing. Have you experimented with different values of K, and how does Dense Backpropagation perform with larger or smaller K values?

2. The paper acknowledges that it doesn't analyze the impact of Dense Backpropagation on fine-tuning MoEs. Do you have any insights or plans to investigate whether the method improves the stability or performance of MoEs during fine-tuning?

---

> ### Author Response · Authors · 2024-11-26
> **Author Response**
>
> Dear Reviewer 3TBL,
> Thank you for your insightful comments, and for highlighting the novelty of our approximation method. In this response, we want to outline how we have addressed your concerns in the revised paper.
>
> > W1: More Evaluation on Tasks
>
> We agree that evaluation beyond perplexity is important. We now present the results of our model on various benchmarks including MathQA, LogiQA, LogiQA 2, MMLU, OpenBookQA, ARC easy, ARC challenge, HellaSwag, COPA, and PiQA, in Figure 1 and Table 1.
>
> > W2: No analysis of Dense Backprop on inference workloads
>
> We have modified our method to only apply the dense approximation in the backward pass. We observed this led to improved results in training and evaluation. This means that at inference, no additional forward computation is necessary. So for inference workloads, our method is identical to the baseline. We have updated the paper to reflect this in lines 198-201.
>
> > W3: Should not mention Attention approximation methods
>
> We agree with your concern on presenting the two additional attention methods. Although they demonstrated strong preliminary results, ultimately we came to the conclusion that they cannot be made efficient enough to justify the improvement. We moved the entire section on our attention to the appendix as optional supplementary material.
>
> > W4: Compare to more recent work in MoE Routing
>
> Thank you for the references. We have updated the paper to compare to more recent work in MoE routing. Our baseline is DeepSeek (we use their fine-grained MoEs) and we use auxiliary loss. GRIN uses SparseMixer, so we have added a comparison to SparseMixer in Figure 8. We reproduce the text here for convenience:
>
> > \citet{liu2023sparsebackpropagationmoetraining, liu2024gringradientinformedmoe} use Sparsemixer, which estimates the true router gradient without straight-through. Our method is distinct in that we also provide gradient updates for the expert weights(as seen in Figure 8). \citet{liu2024gringradientinformedmoe} note that Sparsemixer lags behind TopK~(which our method always outperforms) for the first 0.5T tokens, likely due to the noise that Sparsemixer adds. In the 10B tokens that we were able to run Sparsemixer for, our method significantly outperforms it.
>
> Unfortunately, we have not been able to invest 0.5T tokens into this baseline comparison, but we are currently running this and expect to have it done for the camera ready. By the end of the author response period, we will update this number with a longer run.
>
> > Q1: Ablate K
>
> To answer your question on values of K, we experimented with multiple of K and consistently observed an improvement in training. We report K=4 in Table 2 and K=1 in Table 3, and find that we still outperform the baseline with these other values of K.
>
> > Q2: Finetuning
>
> We are currently working on setting up fine-tuning experiments and will have the results of these by the camera ready. Our intuition is that by allowing the router and experts to receive a dense gradient, we can potentially overcome suboptimal routing decisions that were learned during pretraining.

---

> > ### Author Response · Authors · 2024-12-02
> >
> > Dear Reviewer 3TBL,
> >
> > We would like to ask whether there are any further questions we can answer, because today is the last day for reviewers to ask authors questions. In our initial response we added a table with zero-shot benchmark results, comparisons to Sparsemixer, and ablations over K. We would also like to draw your attention to our response to Reviewer XWKc, where provided results showing that our method is compatible with Sparsemixer and actually improves it. This is because Sparsemixer focuses on the router update, whereas our method can update both the router and the experts with a dense gradient, or some combination thereof.

---

### Official Review · Reviewer_X235 · 2024-11-04

**Soundness:** 3
**Presentation:** 3
**Contribution:** 3
**Rating:** 8
**Confidence:** 4

**Summary:**

The paper provides an improved back propagation method for MoE routing where the back propagation flows through the outputs of the experts that are not among these sparsely activated ones. This is done by approximating the outputs  of non-executed experts as the average output of other similar tokens executed by the expert. Similar tokens may be defined by those routed to a similar set of experts, or those that have a high attention score with the given token. Experiments show that this method gives improved metrics without increasing training compute.

**Strengths:**

Gives a simple method to approximately estimate outputs of each expert for each input to improve the backprop gradients to the router. It is nice to see in Figure 9 that quality of approximation of the router gradient is fairly high.

Does not increase the computation required for the backprop

Experiments show improved metrics.

The method provides better load balance.

The paper is fairly well written.

**Weaknesses:**

Expert may be highly non-linear so averaging in this way need not be a good approximation always. Is there a possibility of a better "higher order" approximation?

The perplexity drop in Table 4 seems small.

**Questions:**

Is there an implicit assumption that the experts are "linear'?

You don’t pass gradients through the expert parameters and are only focused on the router gradients. Could something be done to approximate the expert gradients too?

---

> ### Author Response · Authors · 2024-11-26
>
> Dear Reviewer X235,
>
> Thank you for your generous review. We appreciate you highlighting the quality and efficiency of our method, along with the overall writing of our paper. In this response, we aim to clarify a couple of questions.
>
> > W1: Averaging may not work well when the experts are non-linear
>
> You are correct in mentioning that the experts are non-linear. To answer your question, we are not assuming that the experts will be linear – our assumption for the approximation is that nearby tokens will have a similar expert output and in a large training batch they will span the missing expert output. Additionally, our primary motivation is not to approximate any individual expert output, but instead to approximate the entire batch gradient. We demonstrate the efficacy of this batch approximation in Figure 6. A higher-order approximation is possible, but it would be more computationally expensive and our goal is to minimize overhead
>
> > W2: Perplexity drop in Table 4 is small
>
> The perplexity drop is small, and the average benchmark improvement (in Table 1) is also small. However, we believe the important factor for our method is that it is a *free lunch* so long as the benefit outweighs the overhead. We have carefully documented how the performance improves as we scale up in Table 2, and we find that we improve over the baseline more as we scale up training. And this is scaling in our constrained academic budget, so real deployments will be much larger and we anticipate the improvement of our method over the baseline will be far greater. We also document the overhead scaling in Section 4.4, through Figure 8 and Table 4, and find that for larger models, the overhead decreases. We also note that our main perplexity and benchmark improvement results are without the method variants that we have introduced, and as we see in Table 3, these variants improve our method even more. In summary: *our improvement is marginal at small scale, but grows as we scale up training*.
>
> > Q1: Do we assume the experts are linear?
>
> Our expert group approximation does not assume that the experts are linear. Rather, it assumes that an expert output for a given token is in the span of the expert outputs of other tokens.
>
> > Q2: Could something be done to approximate the expert gradients too?
>
> Our method actually also sends an additional gradient update to experts using this method. This is explained in the new draft of the paper in lines 266-301.

---

> > ### Comment · Reviewer_X235 · 2024-11-26
> >
> > Thanks for the response. I will keep my score.

---

### Meta-Review · Area_Chair_mtDG · 2024-12-19

**Metareview:**

The paper presents a dense gradient update for MoE through a lightweight approximation method, while only sparsely activating its parameters. The method centralizes around the assumption that the output of an inactivated expert can be approximated by the outputs of other tokens routed to that expert. From there, the authors propose an expert group approximation to identify the relevant tokens given the current token and the inactivated experts.

Weaknesses:
Authors have addressed most concerns of the reviewers (demonstrations on more challenging tasks, more rigorous evaluations, editing to tone-down claims etc), yet some concerns remain unaddressed such as: how does the proposed method improve stability or performance of MoEs during fine-tuning? [Reviewer 3TBL]. In response, authors claim that they are running experiments, but the results have not been reported.

However, I remain skeptical about whether and to what extent the STE-estimated dense gradients can effectively guide the sparse forward results. Does the method perform well because it closely approximates STE, or is its success primarily due to increased gradient norms, which might behave similarly to using a higher learning rate? [Reviewer XWKc]

The paper has significant merits to stabilize MoEs and improve its performance, yet does not substantiate the claims adequately. The paper might need more editing before being published.

**Additional Comments On Reviewer Discussion:**

Authors have tried to address most of reviewer's comments and have addressed most, yet some concerns still remain. Although Reviewer XWKc has improved score during the reubttal phase, it comes at a drop of the confidence level.

---

### Decision · Program_Chairs · 2025-01-22

Reject